



# Linking scales of sea ice surface topography: evaluation of ICESat-2 measurements with coincident helicopter laser scanning during MOSAiC

Robert Ricker[1,*], Steven Fons[2,3,*], Arttu Jutila[4], Nils Hutter[5,4], Kyle Duncan[6], Sinead L. Farrell[7,2], Nathan T. Kurtz[3], and Renée Mie Fredensborg Hansen[8, 9, 10]

[1]NORCE Norwegian Research Centre, Tromsø, Norway
[2]Department of Atmospheric and Oceanic Sciences, University of Maryland, College Park, Maryland, USA
[3]Cryospheric Sciences Laboratory, NASA Goddard Space Flight Center, Greenbelt, Maryland, USA
[4]Alfred Wegener Institute, Helmholtz Centre for Polar and Marine Research, Bremerhaven, Germany
[5]Cooperative Institute for Climate, Ocean and Ecosystem Studies, University of Washington, USA
[6]Earth System Science Interdisciplinary Center, University of Maryland, College Park, MD, USA
[7]Department of Geographical Sciences, University of Maryland, College Park, MD, USA
[8]Department of Geodesy and Earth Observation, DTU Space, Elektrovej Building 328, 2800 Kongens Lyngby, Denmark
[9]Department of Civil and Environmental Engineering, NTNU, Gløshaugen - Høgskoleringen 7a, 7491 Trondheim, Norway
[10]Arctic Geophysics, University Centre in Svalbard (UNIS), Longyearbyen, Svalbard, Norway
*These authors contributed equally to this work.

**Correspondence:** Robert Ricker (rori@norceresearch.no)

**Abstract.** Information about the sea ice surface topography and related deformation are crucial for studies of sea ice mass balance, sea ice modeling, and ship navigation through the ice pack. NASA's Ice, Cloud, and land Elevation Satellite-2 (ICESat-2) has been on-orbit for nearly four years, sensing the sea ice surface topography with six laser beams capable of capturing individual features such as pressure ridges. To assess the capabilities and uncertainties of ICESat-2 products, coincident high-resolution measurements of the sea ice surface topography are required. During the year-long Multidisciplinary drifting Observatory for the Study of Arctic Climate (MOSAiC) Expedition in the Arctic Ocean, we successfully carried out a coincident underflight of ICESat-2 with a helicopter-based airborne laser scanner (ALS) achieving an overlap of more than 100 km. Despite the comparably short data set, the high resolution measurements on centimetre scales of the ALS can be used to evaluate the performance of ICESat-2 products. Our goal is to investigate how the sea ice surface roughness and topography is represented in different ICESat-2 products, and how sensitive ICESat-2 products are to leads and small cracks in the ice cover. Here we compare the ALS measurements with the ICESat-2's primary sea ice height product, ATL07, and the high-fidelity surface elevation product developed by the University of Maryland (UMD). By applying a ridge-detection algorithm, we find that 16% (4%) of the number of obstacles in the ALS data set are found using the strong (weak) center beam in ATL07. Significantly higher detection rates of 42% (30%) are achieved when using the UMD product. Only one lead is indicated in ATL07 for the underflight, while the ALS reveals mostly small, narrow and only partly open cracks that appear to be overlooked by ATL07. More research on how even small leads can be detected by ATL07 using additional validation data sets and complementing measurements, such as airborne thermal infrared imaging, would be useful to further improve the sea ice data products.



## 1 Introduction

Sea ice is not a planar surface but appears in a wide range of multifaceted shapes. While level ice is the product of solely
thermodynamic ice growth, mechanical processes produce deformed ice. In the presence of winds and waves, ice floes can
collide with each other and pile up into pressure ridges. These ridges can appear as almost linear features in the sea ice surface
topography. The height of sea ice ridges above the surrounding level ice, known as the sail height, is required for the estimation
of drag coefficients. These drag coefficients indicate the intensity of air-ice interactions in the momentum balance equation
describing the ice motion in sea ice models (Castellani et al., 2014; Mchedlishvili et al., 2022). The geometry of ridges also
plays a role in the distribution of snow on sea ice. Snow is redistributed continuously through winds and accumulates at
obstacles such as pressure ridges (Wagner et al., 2022). Eventually, the deformation of sea ice becomes an important factor for
the sea-ice mass balance (von Albedyll et al., 2022; Ricker et al., 2021).

On the other hand, in case of divergent forces or shear, the ice cover breaks apart leaving open water in the form of cracks
and leads. Their width can vary between a few meters to more than a kilometer. Leads are important for the energy transfer
between ocean and atmosphere, but are also essential for satellite altimetry observations as they are required to calculate the sea
ice freeboard, the height of the ice surface above the water level (Ricker et al., 2014). Detecting and measuring the dimensions
of sea ice surface features like ridges and leads is therefore of high interest in order to improve our understanding of the Arctic
climate system.

Information and precise mapping of sea ice surface topography exist mostly from direct measurements acquired during
field campaigns, ship-based surveys, or ice camps. But retrieving continuous basin-scale information about evolution and
distribution of deformed ice, ridges, and leads is difficult. Satellite altimeters like CryoSat-2 are capable of detecting leads
and measuring freeboard (Wingham et al., 2006), but cannot resolve the surface topography to a level that is required to
measure dimensions of ridges, such as the sail height. However, the development of satellite altimeter sensors is advancing,
and in 2018 the National Aeronautics and Space Administration (NASA) launched the Ice, Cloud and land Elevation Satellite-2
(ICESat-2). ICESat-2 carries the photon-counting Advanced Topographic Laser Altimeter System (ATLAS), which surveys the
ground with six beams, arranged in three pairs, where each beam has a nominal footprint diameter of around 11 m (Magruder
et al., 2020). The small footprint and high-pulse repetition rate allow for unprecedented measurements of the sea ice surface
topography. Kwok et al. (2019a) demonstrated that ICESat-2 is capable of resolving rough surface topography via comparisons
with airborne laser altimetry measurements. Fredensborg Hansen et al. (2021) used the geolocated photon heights from ICESat-
2 to estimate the degree of sea ice ridging in the Bay of Bothnia. Recently, Farrell et al. (2020) developed a high fidelity product,
which optimizes the use of information retrieved by the photon-counting technique to detect individual ridges, leads and melt
ponds, and a recent study by Duncan and Farrell (2022) shows the distribution of pressure ridges on the basin-scale. So far,
these ICESat-2 surface elevation products have been mostly bench-marked against airborne lidar measurements from Operation
IceBridge (OIB) with footprints of 2 m (Kwok et al., 2019a), enough to verify the presence of ridges, leads and melt ponds. Yet,
to capture the exact dimensions of ridges and surface features, validation data of even higher resolution is required. Moreover,



if we want to understand how the photon heights relate to the surface roughness within the illuminated area of the footprints, we need detailed and accurate measurements of the surface topography within the illuminated areas of the beams.

Here we present a new validation data set for ICESat-2 sea ice measurements, which has been acquired during the Multidisciplinary drifting Observatory for the Study of Arctic Climate (MOSAiC) (Nicolaus et al., 2022). The helicopter onboard the drifting research vessel RV *Polarstern* was equipped with an airborne laser scanner (ALS) capable of sensing the sea ice surface with a lateral resolution of a few centimetres (Jutila et al., 2022). On 23 March 2020, we followed an ICESat-2 ground track in close vicinity for 130 km, achieving an overlap between the ALS swath and the center beam pair of about 90 %. Although this data set only includes one coincident helicopter flight, we will show that even with a short data set, a comprehensive verification of ICESat-2 sea ice surface elevation products is possible. ALS surveys have been carried out during the entire MOSAiC drift, providing a unique data set of sea ice surface topography through a full seasonal cycle. This study will link the MOSAiC ALS measurements with ICESat-2 measurements, to investigate the evolution of surface topography and deformation of the sea ice near the MOSAiC camp in the context of regional and Arctic-wide changes captured by ICESat-2. We pursue the following goals: First, we aim to validate the ICESat-2 ATL07 (Sea Ice Heights, Level 3A) product (Kwok et al., 2021a), which contains along-track heights for sea ice relative to the WGS84 ellipsoid as well as parameters useful for the detection of open water leads, such as the return and background photon rates. For comparison, we will use the high resolution ALS surface elevations as well as the ALS reflectance, which is used to detect leads. Second, we seek to investigate how the surface roughness within the ATL07 segments/footprint is related to the height estimates. Third, we aim to quantify to which degree the true dimensions of sea ice surface topography, given by the ALS, can be captured by ICESat-2 products. Therefore we will use the official NASA release ATL07 product, but also the high-fidelity product provided by the University of Maryland (Farrell et al., 2020; Duncan and Farrell, 2022), denoted UMD-RDA hereafter. And fourth, we will compare the weak and strong beam with regard to the objectives mentioned above.

Another aim of this study is to demonstrate that with increasing resolution of satellite altimeters, validation strategies also need to be adapted. In fact, being able to relate individual surface features to altimeter signals allows also for smaller, more flexible and less extensive campaigns, in addition to the large scale campaigns that cover different ice regimes and regions. Mapping the dimensions of sea ice surface topography requires validation with high resolution sensors, contrasting the validation for previous altimeters, e.g. CryoSat-2, that was primarily based on comparing large-scale averages.

## 2 Methods and data

### 2.1 Flight operations and airborne laser scanner (ALS) data

Measurements of sea-ice surface elevation were carried out using the near-infrared (1064 nm), line-scanning Riegl VQ-580 ALS installed in the rear baggage compartment of the helicopter. Moreover, the scientific instrumentation for this helicopter flight contained a Global Navigation Satellite System (GNSS) inertial system Applanix AP 60-AIR. The take off from the RV *Polarstern* flight deck was at 10:37 UTC on 23 March 2020, with clear visibility and no clouds. In the vicinity of the vessel, instruments were switched on and initialized, before intercepting the ICESat-2 ground track. The center beam pair ground track



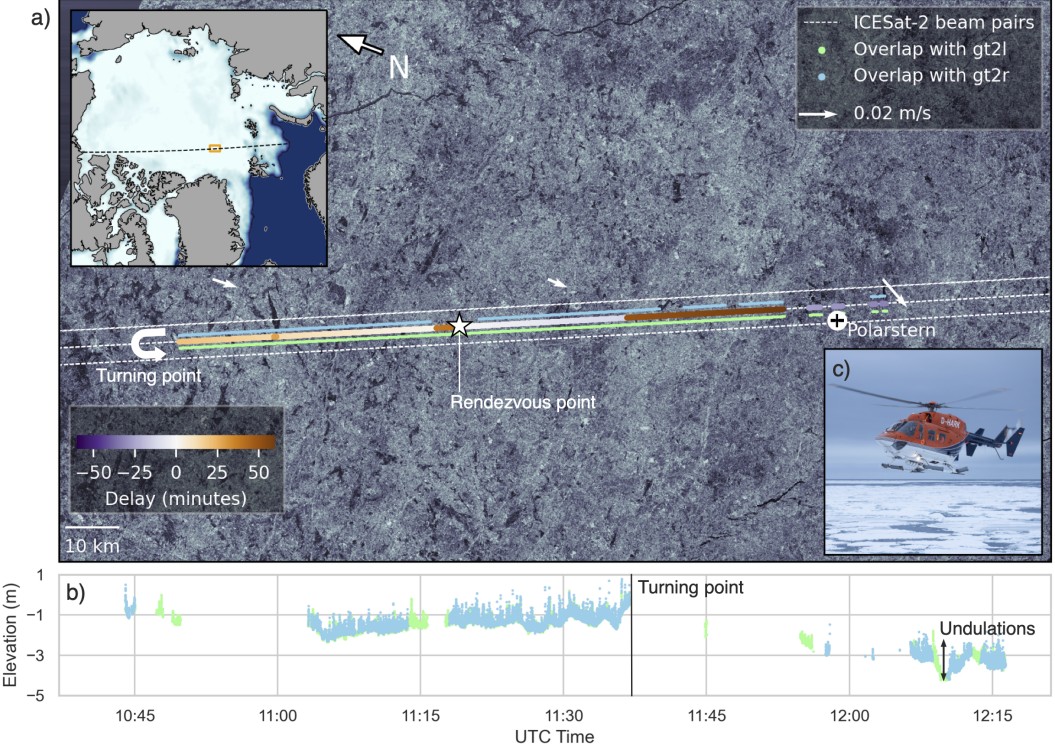

**Figure 1.** a) Overview of helicopter flight and airborne laser scanner measurements (ALS) coinciding with the Ice, Cloud and land Elevation Satellite-2 (ICESat-2) center beam pair, comprising the weak (gt2l) and strong beam (gt2r). The upper left box shows the location of the zoom-in position as an orange rectangle. The time delay of ALS data acquisition is shown for the overlapping sections. Note that part of the overlap has been achieved on the return flight to RV *Polarstern*. The position of RV *Polarstern* corresponds to the time of the helicopter take-off. The background shows a Sentinel-1 radar image at the day of the flight, obtained from Drift&Noise FRAM-Sat (https://framsat.driftnoise.com/) in the framework of the MOSAiC project. White arrows show the low resolution sea ice drift from OSI SAF. b) Elevations of the overlapping ALS measurements along the helicopter flight track before correction of undulations, relative to the DTU21 MSS. Note that gt2l elevation profile is partly masked by gt2r. Also note that on the return section after the turning point, only those sections are shown where overlap haven't been achieved on the outbound flight. c) Helicopter used for the survey (Credits: Alfred-Wegener-Institut / Jan Rohde (CC-BY 4.0)).

was chased towards the North-East (Fig. 1a). At 11:17 UTC, after 40 minutes flight time, the helicopter was passed overhead
by ICESat-2 at the rendezvous point (Fig. 1a). After approximately 130 km, the helicopter returned the same way back to the
ship, in order to close possible gaps in the overlap. Figure 1a shows the overlap of the ALS swath and the strong and weak
ICESat-2 beams, as well as the delay between ALS and ICESat-2 observations. Gaps in the overlap during the outbound flight
could be partly closed on the return flight (Fig. 1b). The survey ended at 12:24 UTC, when the helicopter landed. Considering
the along-track overlap, we achieved a coverage of 97 km for the strong beam and 117 km for the weak beam.

With the aid of the position and altitude data collected by the GNSS inertial system integrated to the sensor, the range
measurements from the ALS were converted into geolocated surface elevation point clouds and referenced to the Technical



University of Denmark mean sea surface (DTU21 MSS) (Andersen, 2022). The elevation point clouds were then filtered to remove atmospheric backscatter, interpolated onto a regular grid with a resolution of 0.50 m, and split into segments of 30 seconds duration. Additional parameters included range corrected reflectance and echo width. With a 60° field of view of the

ALS, the resulting swath width was approximately equal to the nominal flight altitude of 1000 ft (≈300 m) above ground. Leads are detected automatically by drops in reflectance, typically below -7 dB. A more detailed description of the ALS data and their processing can be found in Hutter et al. (2022). The 30 seconds gridded segments are used for co-registration with ICESat-2 measurements.

## 2.2 ICESat-2 data

ICESat-2's ATLAS instrument emits pulses of green laser light (532 nm) that illuminate footprints of around 11 m in diameter on the surface (Magruder et al., 2020). These pulses are repeated at 10 kHz, resulting in over-sampled coverage of one footprint every ∼70 cm. A distinguishing feature of ATLAS is its six beams, separated into three beam pairs, with each pair containing a weak and a strong beam. Beam pairs are separated by 3.3 km, while the beams within a pair are separated by about 90 m (Markus et al., 2017). In this underflight, the helicopter flew beneath the central beam pair of ICESat-2. The strong beam

(hereafter referred to as 'gt2r') was situated on the right side of the direction of spacecraft motion, while the weak beam ('gt2l') was situated on the left side.

The following subsections describe the ICESat-2 data products used in this study.

### 2.2.1 ATL07 product

The primary sea ice elevation product from ICESat-2, ATL07, provides along-track sea ice and sea surface height measurements
at variable length segments for each of the six ground tracks. ATL07 is derived from the ATL03 product, which provides geolocated photon heights, time-varying geophysical corrections, range corrections, and background rates. Moreover, ATL07 uses the atmospheric product ATL09, which provides cloud statistics, backscatter, background rates, and surface atmospheric variables. Each ATL07 segment consists of 150 aggregated signal photons, varying in length from ∼15 to ∼30 m or more. Segment lengths are typically shorter when signal strengths are high (e.g. from specular surfaces or shots from the strong
beams) and longer when signal strengths are weak (e.g. from more diffuse surfaces or shots from the weak beams).

The 150-signal-photon-heights are binned and the resulting histogram is trimmed to remove anomalous values. The trimmed histogram is fit using a dual-Gaussian mixture distribution, and following the procedure in Kwok et al. (2019b, 2022). Then, the surface height is estimated from the fitted distribution. The resultant surface heights are referenced to a blended CryoSat-2/DTU13 MSS, and are provided for both the weak and strong beams (Kwok et al., 2022, 2020). In addition to the surface
heights, the ATL07 product also provides statistics of photons that have been used for the 150-signal-photon-height aggregation. The hist_w parameter provides an estimate of segment height histogram width.

Surface types are classified in ATL07 using the photon rates, fit distribution width, and background rate, and are used to indicate lead points in ATL07. These lead points are necessary to estimate the reference sea surface height and calculate sea ice freeboard, which is done in the ATL10 product (Petty et al., 2020). Due to the lack of suitable leads during this underflight, and





since freeboard validation has been carried out in Kwok et al. (2019a), freeboards from ALS and ATL10 are not considered in this study. For this work, we use the latest available ATL07 version 5 (Kwok et al., 2021a).

### 2.2.2   University of Maryland (UMD) product

The University of Maryland-Ridge Detection Algorithm (UMD-RDA) is a surface retracker for analyzing ICESat-2 altimeter data (Duncan and Farrell, 2022). When applied to the ICESat-2 ATL03 global geolocated photon heights it is used to extract sea

ice height on a per-shot basis. To reduce the impact of background (noise) photons, a photon height distribution using a 5-shot aggregate (~2.8 m along-track distance) is constructed and only photon height estimates within the 15th and 85th percentiles of the distribution are retained. The 99th percentile of the trimmed height distribution is associated with the first interface encountered by the laser and thus defines sea ice surface height. UMD-RDA sea ice surface height estimates are processed wherever ATL07 sea ice heights exist (Kwok et al., 2022), thereby eliminating any cloud-contaminated photon retrievals in

the UMD-RDA estimates. Height corrections are applied for atmospheric range delay, tides, and the mean sea surface (MSS). UMD-RDA surface height is reported relative to DTU18 MSS model (Andersen et al., 2018). The UMD-RDA resolves the sail height of individual pressure ridges on the ice surface and therefore provides a more complete estimate of the height distribution in areas of high surface roughness compared to ATL07 data (Duncan and Farrell, 2022). Recently, UMD-RDA has been applied to ATL03 data collected across the Arctic Ocean and used to investigate sea ice surface roughness, sail height,

ridge width and spacing, and ridging intensity at the end of winter between 2019 and 2022 (Duncan and Farrell, 2022).

### 2.3   Co-registration of ICESat-2 and helicopter laser scanner measurements

The co-registration of ICESat-2 and ALS measurements is based on the segments defined in the ATL07 product. With the segment length given in ATL07, together with the segment center point location, we construct polygons for each segment. The length of the polygons is the ATL07 segment length, while for the width we choose 13 m. We assume a 13 m diameter as a

conservative estimate and a balance between the pre-launch footprint diameter estimate of 17 m (Markus et al., 2017; Kwok et al., 2019b) and the calculated footprint of around 11 m found in Magruder et al. (2020). Additionally, the 11 m footprint in Magruder et al. (2020) covers the $1/e^2$ diameter, whereas assuming 13 m allows us to capture the full beam diameter and also account for geolocation uncertainties (Luthcke et al., 2021). The polygon edges are rounded, as we assume a circular footprint. In the next step, we collect all ALS measurements within the boundary of each polygon and assign them to the corresponding

segment. Arithmetic average and standard deviation of all ALS measurements within each segment are calculated. At this point we apply a correlation-based drift correction that is described in section 2.4. The co-registration and drift correction are carried out for each ALS section (30 seconds length). Finally, all segments with co-registered ATL07 and drift-corrected ALS measurements are concatenated. In the following we refer to the co-registered ATL07 as ATL07 seg, and to the co-registered and segment-averaged ALS measurements as ALS seg. We also keep all the ALS elevation points registered for individual

segments and refer to them as ALS full in the following.

     Figure 2 shows an example of a 1 km profile section, where the two ICESat-2 center beams gt2r and gt2l are overlapping with the laser scanner swath. The zoom-in figure shows the ATL07 segment polygons of gt2r, which are overlapping each





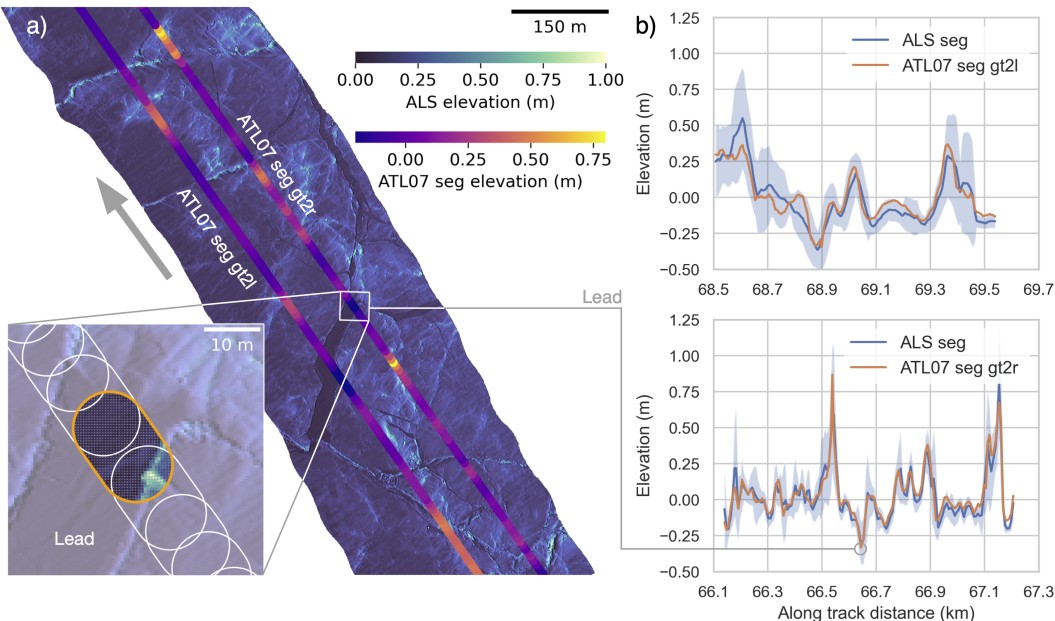

**Figure 2.** a) Profile section (1 km) of the airborne laser scanner (ALS) swath with overlapping ICESat-2 beams after drift correction. The zoom-in box shows the ICESat-2 segment outlines in white and one segment highlighted in orange, with matching ALS point measurements inside, corresponding to a lead. The grey arrow indicates the flight direction of the helicopter. b) Elevation profiles (along the overlap shown in a)) of ICESat-2 beams and coincident ALS elevation, averaged within the corresponding ICESat-2 segments. The blue shaded area in the elevation profiles represents the standard deviation of ALS point measurements within segments.

other, with a typical distance of 6-7 m between the segment centroids. For one of the segments, the assigned laser scanner points are shown.

If ATL07 segments are covered twice, on the outbound as well as on the return flight, the outbound flight is prioritized and ALS measurements from the return flight are not considered for co-registration. We calculate an along-track distance, which starts at zero at the time when the first overlap between ALS and ATL07 is registered. The along-track distance is continued also after the turning point. Therefore, ATL07 segments that are only covered during the return flight appear towards the end of the profile when referring to the along-track distance. Figure 1b shows co-registered ALS points after the turning point that

fill a gap at the beginning of the outbound flight.

    The UMD-RDA product is included into the co-registration of ATL07 and ALS by evaluating the timestamps of UMD-RDA and co-registered ATL07.

## 2.4 Drift correction

Due to the time difference between ALS and ICESat-2 observation, the co-registration of ICESat-2 and ALS measurements

is affected by the drift of the sea ice. From observations at *Polarstern* and evaluation of the Ocean and Sea Ice Satellite





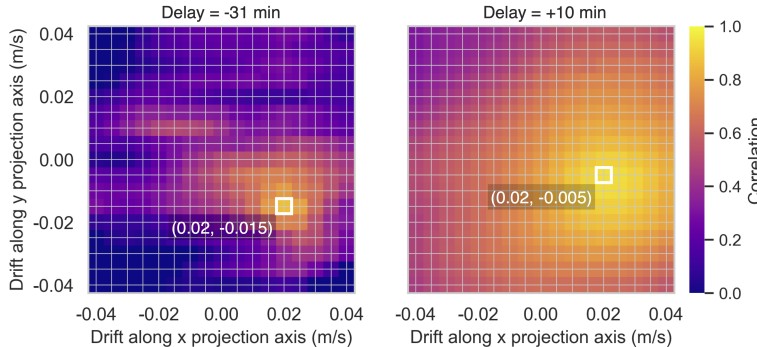

**Figure 3.** Evaluation of sea ice drift correction for two different airborne laser scanner (ALS) data frames from 31 minutes before the rendezvous point, and 10 minutes after. Correlation between ALS and ICESat-2 segment elevations for applied sea ice drift in steps of 0.005 m s$^{-1}$ in x and y direction of the polar stereographic projection. White squares highlight sea ice drift in x and y direction, where the maximum correlation between ALS and ICESat-2 segment elevations is achieved.

Application Facility (OSI SAF) low-resolution ice drift product (Lavergne et al., 2010), we know that ice drift was rather weak in the region, with magnitudes of 0.01 - 0.02 m s$^{-1}$ in southerly direction (Fig 1a). But even under such conditions, expected offsets will be up to ~70 m, assuming the maximum delay of about 60 minutes. Considering the sampling rates and spatial resolution of ALS and ICESat-2 (Sections 2.2 and 2.2) shows the need for ice drift correction. To evaluate and correct the effect

of ice drift in between helicopter and satellite data acquisitions, we calculate correlations between ATL07 segment elevations and segment-averaged ALS elevations after incrementally applying drift corrections to the original ALS point measurements before co-registration in steps of 0.005 m s$^{-1}$ in both x and y direction of the polar stereographic projection axes. Figure 3 shows two examples of correlation coefficients between ATL07 segment elevations and segment-averaged ALS elevations after applying a set of a priori assumed drift components on an ALS data frame. Here we assume a range of sea ice drift velocities

between -0.04 m s$^{-1}$ to 0.04 m s$^{-1}$ in each direction. The correlation analysis shows that in case of a 30 seconds data frame from 31 minutes before the rendezvous point, the maximum correlation of 0.87 is reached with sea ice drift 0.02 m s$^{-1}$ in x direction and -0.015 m s$^{-1}$ in y direction. Similarly, for the second example, where correlations are generally higher, we find the maximum correlation of 0.92 with sea ice drift of 0.02 m s$^{-1}$ in x direction and -0.005 m s$^{-1}$ in y direction. With this method applied to every 30 seconds ALS data frame, we find sea ice drift velocities ranging between 0.014 and 0.02 m s$^{-1}$,

which corresponds well with the OSI SAF low resolution drift product (Fig 1a). Due to the choice of 0.005 m s$^{-1}$ binning, we expect uncertainties in x and y directions of at least 0.0025 m s$^{-1}$.

### 2.5 Removal of undulations due to poor GPS-INS solution in ALS data

Mainly due to the poor GPS solution close to the North Pole, the entire ALS profile reveals undesired undulations and erroneous gradients in the along track direction in the ALS elevations of partly several meters in magnitude (Fig. 1b). These need to be

corrected. We first detect gaps in the elevation profile, caused by missing overlap between ALS and ICESat-2, and then divide



the profile into segments that are free of gaps. To correct each profile segment for erroneous gradients, we apply a moving Gaussian window of 5 km length along each profile segment and remove the obtained low-pass filtered signal from the original data. The choice of the 5 km window length is supported from findings in Hutter et al. (2022), showing that most of the variability is caused by undulations at scales of 5km and larger. Finally, we concatenate the individual profile segments to

receive the corrected ALS elevation profile. We note that this correction will also eliminate natural long-wave signals and gradients from the ALS profile, and freeboard estimation will be corrupted. But since for this study we only consider the surface topography, removal of long-wave signals does not affect our analysis. For conformity, this correction is also applied to ICESat-2 data, including ATL07 seg and UMD-RDA. Consequently, all elevation products after this correction are referenced to the low-pass filtered surface elevation.

## 2.6   Mapping of obstacles and their dimensions

To evaluate how good ATL07 seg and UMD-RDA are able to map the surface topography and dimensions of features, we apply a peak detection algorithm. Our method is based on the function *find_peaks* as part of the *python* signal processing library *scipy.signal*. The function locates local maxima and calculates the height of the peak, which here is defined as the height of an obstacle above the local level sea ice. From a given peak, we evaluate if a virtual horizontal line intersects the

slope of another peak on the left or right within a given maximum distance of 250 m to either side. This value has proved reasonable after empirical evaluation using different values. Either within the maximum distance, or until the next peak, we search for the minimum on either side of the peak. The higher value then represents the elevation of the local level sea ice. The height of an obstacle is calculated as the vertical difference between the peak elevation and the elevation of the local level ice. To be classified as an obstacle, this difference must exceed 0.6 m (Hibler III et al., 1972; Duncan et al., 2018). The

minimum distance between two neighbouring peaks is set to 16 m, which is approximately two times the distance between the centroids of two subsequent ATL07 segments. This method also complies with the Rayleigh criteria: two maxima points must be separated by a point with a height smaller than half of the maxima to be resolved as separate features (Hibler, 1975; Castellani et al., 2014). Here we apply a more strict criteria in case of high-resolution data, for example in case of two peaks, separated by a point with lower elevation, fulfilling the Rayleigh criteria as described above. When using high-resolution data,

like ALS full or UMD-RDA, with point spacing < 1 m, we assume that these two peaks belong to the same surface feature, and with our method, only one would be registered.

We also estimate the widths of obstacles. The height of the obstacle is halved and subtracted from the peak elevation. At the resulting height a horizontal line is drawn and the width of an obstacle is then given by the distance between the intersections of the line with the slope on either side of the peak. The minimum width is given by 1 m. The spacing of obstacles is given by

the along track distances between subsequent obstacles.





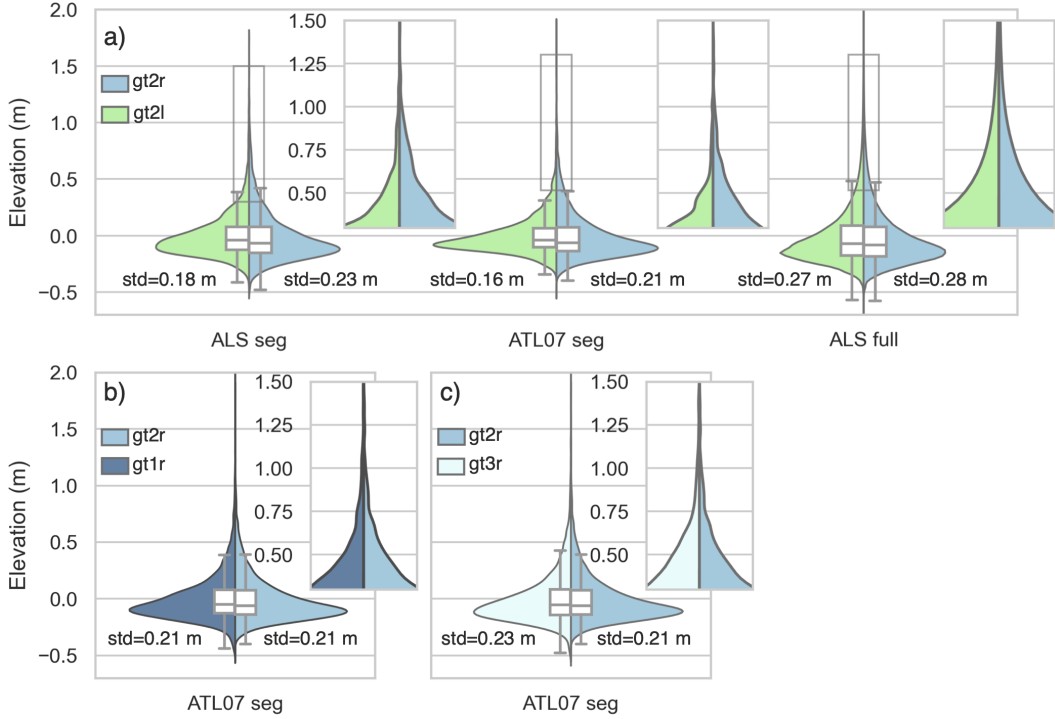

**Figure 4.** a) Violin and box plots showing the elevation distributions from ATL07 segments (ATL07 seg), the segment-averaged ALS elevations (ALS seg), and the ALS elevations from all co-registered points at 50 cm resolution (ALS full), separated between weak (gt2l) and strong (gt2r) beam segments. b) Violin and box plots showing the elevation distribution of the strong beams of beam pair 2 and 1. c) Violin and box plots showing the elevation distribution of the strong beams of beam pair 2 and 3.

## 3 Results

### 3.1 Comparison between ATL07 and ALS co-registered elevations

For the entire profile of co-registered data, Fig. 4a visualizes the height distributions, divided into the weak beam gt2l, and the strong beam gt2r. Here we consider three different data sets: ALS seg, ATL07 seg, and ALS full, as introduced in section

2.3. All three data sets reveal a log-normal like distribution with a long tail towards higher elevations. Comparing strong and weak beams, we find that the gt2r (strong beam) distribution for both ALS seg and ATL07 seg show a higher dynamic range than gt2l, indicated by the standard deviations of 0.23 m for ALS seg (gt2r) and 0.21 m for ATL07 seg (gt2r), compared to 0.18 m for ALS seg (gt2l) and 0.16 m for ATL07 seg (gt2l). Moreover, the gt2r distributions contain a larger fraction of high elevations. ALS seg (gt2r) and ATL07 seg (gt2r) contain 3.8 % and 3.3 % of elevations > 0.5 m, while ALS seg (gt2l) and

ATL07 seg (gt2l) only contain 2.0 % and 1.4 % of elevations > 0.5 m.





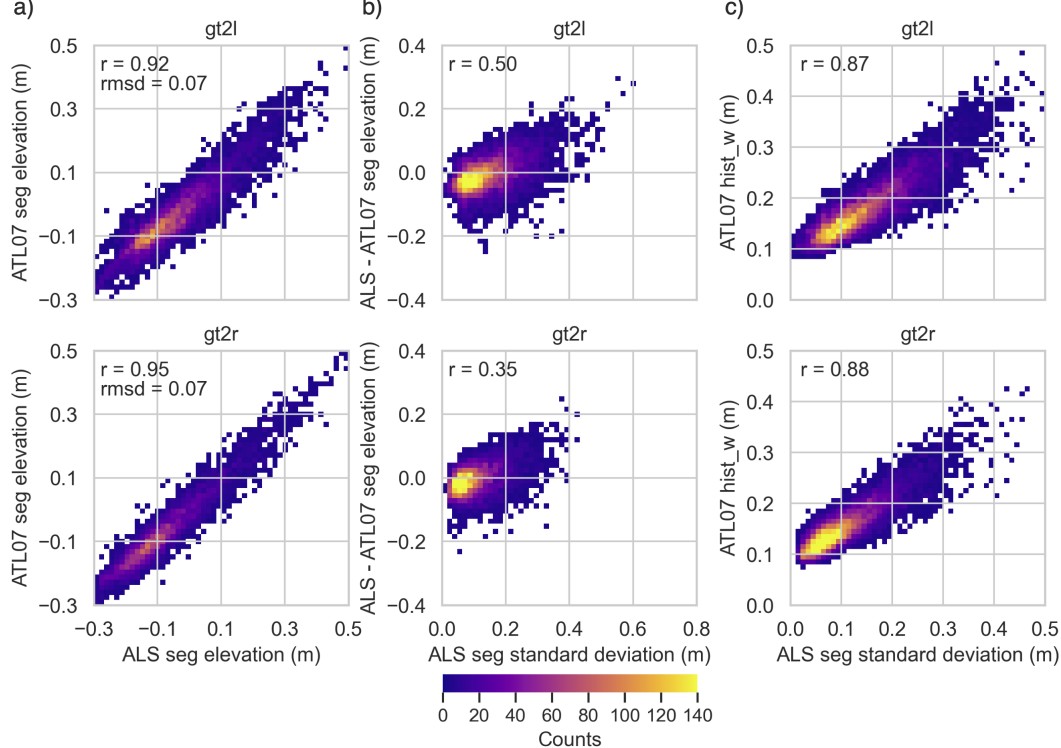

**Figure 5.** Scatter plots of co-registered ALS and ATL07 elevations: a) Comparison of segment-averaged elevations with given root-mean-square deviation (rmsd) and Pearson correlation coefficient (r), b) Comparison between ALS segment standard deviation and difference between ALS and ATL07 segment elevations, and c) comparison between ALS segment standard deviation and ATL07 hist_w, representing the width of the height distribution within segments, provided in the ATL07 data product.

Considering ALS full, the distribution reveals significantly higher fraction of elevations > 0.5 m (5.3 % for gt2r) compared to ALS seg and ATL07 seg, and generally a higher dynamic range with standard deviations of 0.28 m and 0.27 m for gt2r and gt2l. In contrast to ALS seg and ATL07 seg, gt2l and gt2r distributions of ALS full reveal a similar dynamic range.

To illustrate how representative the results in this study are of the surrounding ice conditions, we compare gt2r with the statistics for the outer strong beams gt1r (Fig. 4b) and gt3r (Fig. 4c). We find similar elevation distributions with standard deviations of 0.21 m and 0.23 m for gt1r and gt3r, compared to 0.21 m for gt2r.

Considering the along-track signal, Fig. 2 shows the elevations of ALS seg and ATL07 seg along a 1 km profile section. In general, ALS seg and ATL07 seg reveal a similar variability and good agreement even on small scales, but the gt2r profiles reveal a higher dynamic range, higher amplitudes and generally more variations than the gt2l profiles. The agreement between ALS seg and ATL07 seg is also shown in Fig. 5a, which presents a comparison between all co-registered elevations of the entire helicopter flight. The correlations between the segment-averaged ALS elevations and corresponding ATL07 elevations are 0.92 for gt2l, and 0.95 for gt2r. Root mean square deviations (rmsd) are 0.07 m for both gt2l and gt2r. Here, we acknowledge that





the value of the rmsd is limited by the fact that we have subtracted a long wave signal from the elevation data sets, and therefore the rmsd does not relate to the heights of the original data sets.

The relationship between surface roughness and retrieved segment-scale elevations is assessed in Fig. 5b, we consider the standard deviations for all ALS elevation points within each segment and compare them with the elevation difference between ALS seg and ATL07 seg. This indicates how the ATL07 product is affected by surface roughness within the segments. For both beams, the difference distributions are slightly skewed, with correlations of 0.5 and 0.35 for gt2l and gt2r beams, respectively.

Figure 5c compares the standard deviations for all ALS elevation points within each segment with ATL07 hist_w from the
individual photons within each ATL07 segment, given in the ATL07 product. Here we find a correlation of 0.87 for gt2l and 0.88 for gt2r. This shows that the individual photon heights that are used to calculate the ATL07 segment heights can reproduce the actual surface roughness within the segments to a certain degree. We will see in section 3.4 that an advanced evaluation of individual photon heights derived from ATL03 (ATL UMD) is capable of capturing high fidelity surface topography features on meter scale.

In the following section, we investigate in more detail how surface roughness is represented in the ATL07 segments.

## 3.2   Comparison between ATL07 and ALS within segments

This section utilizes the high-resolution photon elevations that make up each ATL07 segment to compare with ALS elevations within these segments.

Figure 6 shows four examples of ATL07 segments and the photon height distributions that make up each segment, as
well as the corresponding ALS elevations from within the same segment. To compare the two datasets in light of sampling discrepancies (e.g. differences in sensor frequency and measurement counts within each segment), a lognormal distribution was fit to both datasets and statistics such as the mean, mode, and standard deviation were drawn from these fit distributions. Over relatively smooth ice, the means and standard deviations of the fit distributions from ATL and ALS agree to within one cm across both beams. Additionally, the lognormal fit to the ATL photons shows a standard deviation also within one cm of
the hist_w, corroborating the roughness estimate given in the ATL07 product. Over rough ice, the within-segment agreement worsens, though the shapes of the distributions remain similar. Mean within-segment elevations differences are around 10 cm (30 cm) for the strong (weak) beam, while standard deviations differ between around 13 cm (strong beam) up to 80 cm (weak beam). As seen in the smooth ice examples, the standard deviations of the lognormal fits agree well with the given hist_w for both beams.

While the examples shown in Fig. 6 demonstrate relatively good agreement between the ALS within-segment roughness (standard deviation of elevation measurements) and the ATL07 within-segment roughness (standard deviation of photon heights), this is not always the case across all overlapping segments. Figure 7 shows the distribution of roughness estimates in all segments, from ALS and ATL07, as well as the ATL07 hist_w. It is clearly seen that using the ATL photons to estimate roughness can miss the extremely smooth (<10 cm) and extremely rough (>40 cm) ice, and instead result in more moderate
(10-30 cm) roughness values. It is important to note that the differing system impulse responses from ALS and ICESat-2 are





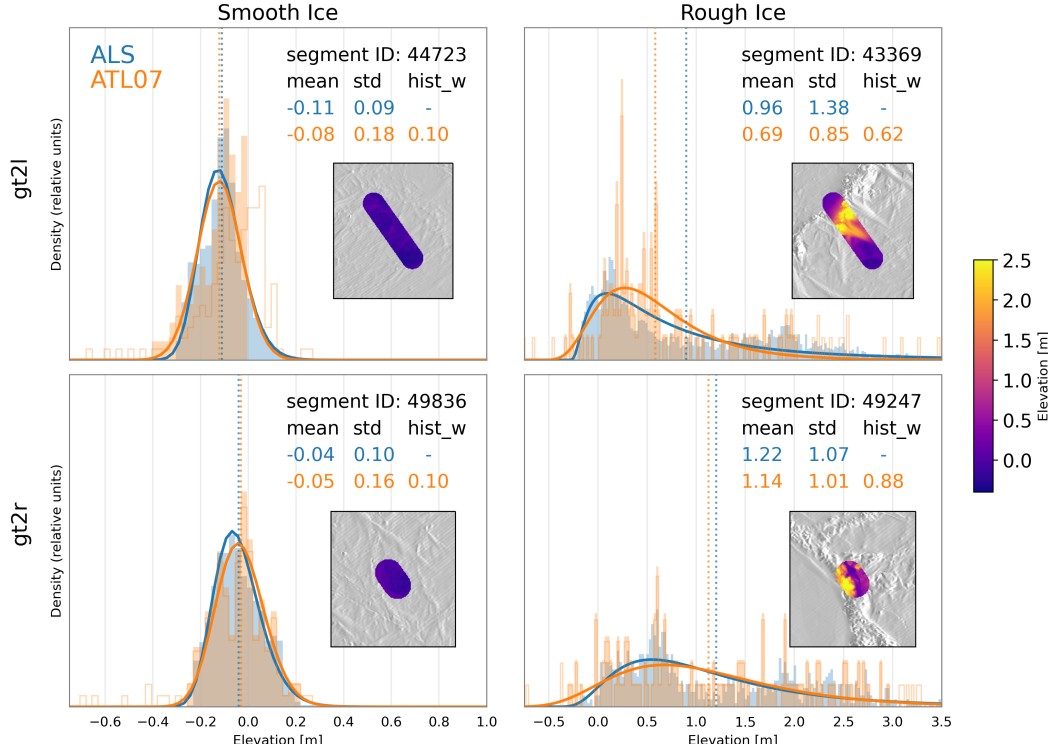

**Figure 6.** Example ATL07 segments and photon heights that comprise each segment, compared with ALS within-segment elevations. Example segments are chosen to represent relatively smooth ice (left) and rough ice (right) from the weak beam gt2l (top) and strong beam gt2r (bottom). The orange histograms (bars) show the elevation distributions from the full, untrimmed photon histogram (unfilled bars) as well as the trimmed photon histogram (filled bars) that makes up the ATL07 segment. The associated within-segment ALS elevations are shown as filled blue bars. Bin sizes are 2.5 cm for all examples. The trimmed ATL07 and ALS histograms are fit with a lognormal distribution (solid lines), from which the provided mean and standard deviations (in meters) are drawn. The hist_w represents the width of the height distribution provided in the ATL07 data product. Inset plots show the within-segment ALS elevations highlighted on top of the surrounding ALS ice elevations (shown in hillshade).

not accounted for in this analysis, which likely explains the observed differences over very smooth ice. Further explanation behind these differences and their potential impacts are discussed in Section 4.3.

### 3.3 Occurrence of leads in ATL07 and ALS

During this underflight of ICESat-2 by ALS, no leads were identified in the ATL07 weak beam gt2l data product, and only 280 one lead was identified in the ATL07 strong beam gt2r data product. Figure 8 shows this lead detected from beam gt2r as well as the leads detected from ALS along a seven-plus km section of the profile. In addition to the location of the leads, the ATL surface height profile (Fig. 8a) and one lead characteristic parameter for each sensor is given: the photon rate from ATL07 (Fig.





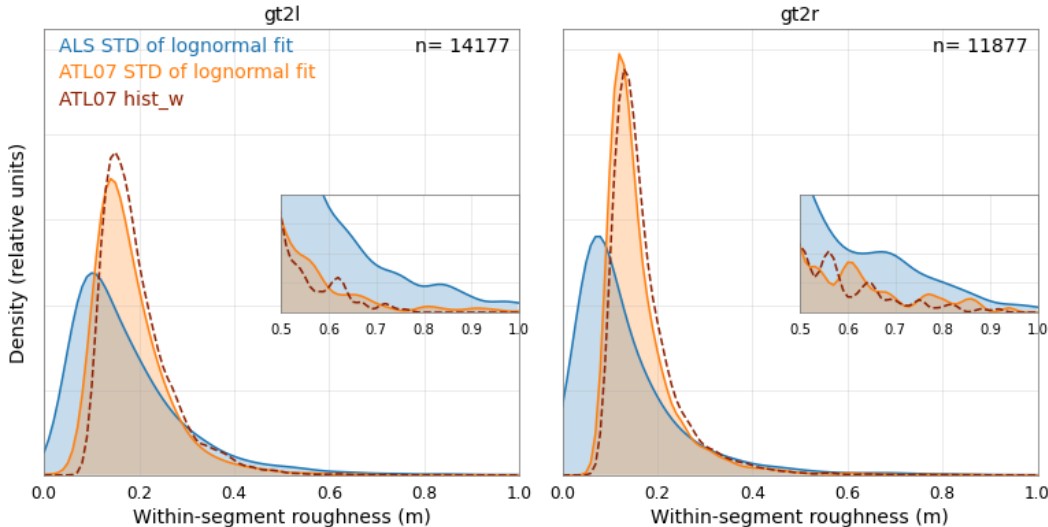

**Figure 7.** Probability density functions (PDFs) of roughness estimates from all overlapping segments for gt2l (left) and gt2r (right). Filled curves show the standard deviations of the lognormal fits to the within-segment elevation histograms from ALS (blue) and ATL (orange). The dashed red curve shows the ATL07 hist_w parameter provided in the data product. Inset axes highlight roughness values between 0.5 and 1.0 m.

8b) and the surface reflectance from ALS (Fig. 8c). These parameters are used in the respective lead detection algorithms, and are shown here to help to identify potential leads that are missed in the classification in the ATL07 product.

The lone lead detected in the ATL07 gt2r data occurs near 239.75 km along-track and can be corroborated by the co-located height minimum and the local maximum in photon rate. The narrow drop in ATL07 heights suggests a relatively small, specular lead. The ALS also records a lead in the same location, indicated through a drop in the surface reflectance. The ALS elevation model (Fig. 8d) and the ALS surface reflectance ( 8e) show a system of narrow, partly open, cracks within older and larger, refrozen leads in the vicinity of the detected lead in ATL07. In total, ALS detects 10 leads over this profile section. Some of

the leads detected from ALS appear to be missed in ATL07, for example occurring at about the 243.5 and 244 km marks in the profile. These locations both show a relative minimum in the surface height and a minimum in the ALS surface reflectance indicative of a lead, though one shows a small peak in the ATL07 photon rate while the other shows a local minimum. Possible explanations for these missed leads are discussed in Section 4.2.

## 3.4 Detection of obstacles using different ICESat-2 products

In this section, we investigate the capabilities of ATL07 seg and the UMD-RDA high fidelity product to detect ridges and measure their sail heights. We use ALS full as the reference, since it provides a point spacing of 50 cm, assumed to represent the true surface topography along the ICESat-2 beams. In addition, we will also consider ALS seg. We use the term 'obstacle' for topographic features including pressure ridges, but also fragments in rubble ice fields. Figure 9a shows a 500 m profile



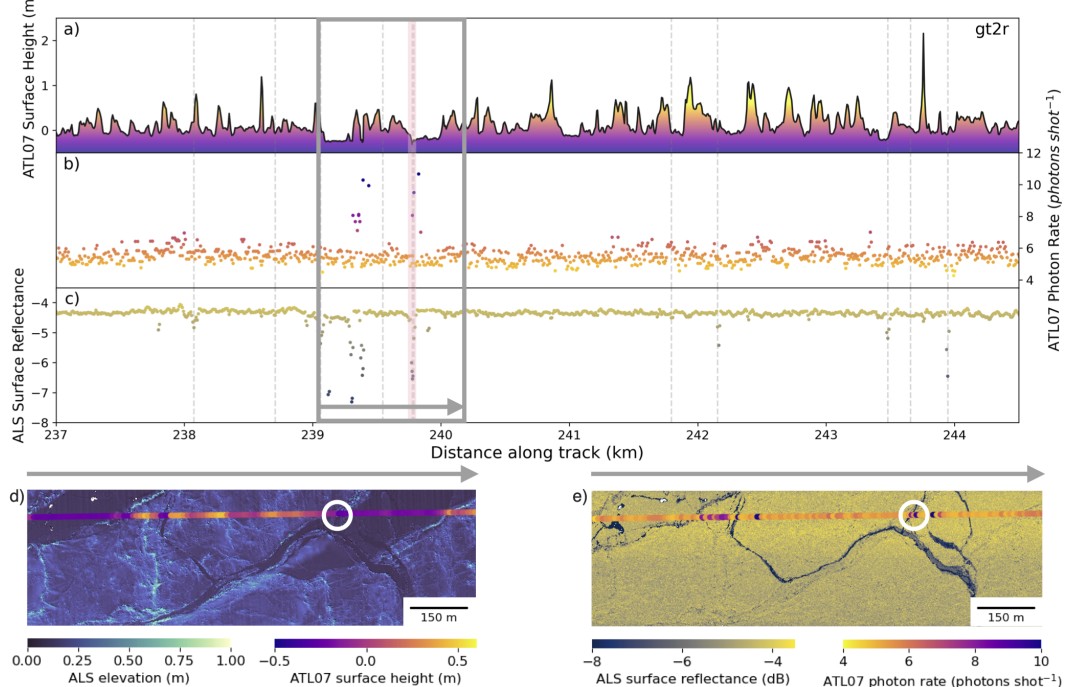

**Figure 8.** Seven-plus km section of the overlapping flight path with leads detected from ATL07 gt2r (vertical pink line) and from the ALS (vertical dashed grey lines). The three line profiles show ATL07 height profile colored by elevation (a), ATL07 photon rates (b), and ALS surface reflectance (c). For b) and c), darker points represent values more probable to be classified as leads. The grey outlined box indicates the length of the subsections showing the gridded ALS elevations and coincident ATL07 gt2r elevations (d), as well as gridded ALS surface reflectance and coincident ATL07 photon rates (e). The grey arrow indicates the flight direction. white circles highlight the only registered lead in ATL07.

section for the strong beam gt2r. Here, elevations are referenced to the level ice. This profile section contains flat level ice, but
also deformed sea ice with elevations up to 1.5 m. While ALS seg and ATL07 seg show a smooth elevation profile, resulting in a reduced dynamic range and in missing peaks, UMD-RDA can also resolve steep slopes and smaller details of the surface, e.g. at 89.25 km (Fig. 9a). Yet, also UMD-RDA cannot resolve all the peak heights given by ALS full. Within the 500 m section, seven obstacles with heights of 0.6 m above the local level ice are detected in ALS full. On the other hand, we find three obstacles in UMD-RDA and only one in ATL07 seg and ALS seg. Figure 9b shows the statistics of obstacle detection for the
entire flight profile. In total 532 obstacles are detected within ALS full, 225 in UMD-RDA, 87 in ATL07 seg, and 102 in ALS seg. The shape of the density distribution of detected obstacle heights is similar among all products, but heights > 1.8 m are sparse in ATL07 seg and ALS seg. The distribution of obstacle widths reveals a mean of 7.7 m for ALS full, while for ATL07 seg and ALS seg, average widths are 39.6 and 38.4 m. The widths of the UMD-RDA obstacles are in between these values (24.4 m). The spacing between obstacles is a consequence of the detection counts, and therefore we find the lowest spacing for
ALS full, while ATL07 seg and ATL07 seg show the highest spacing.




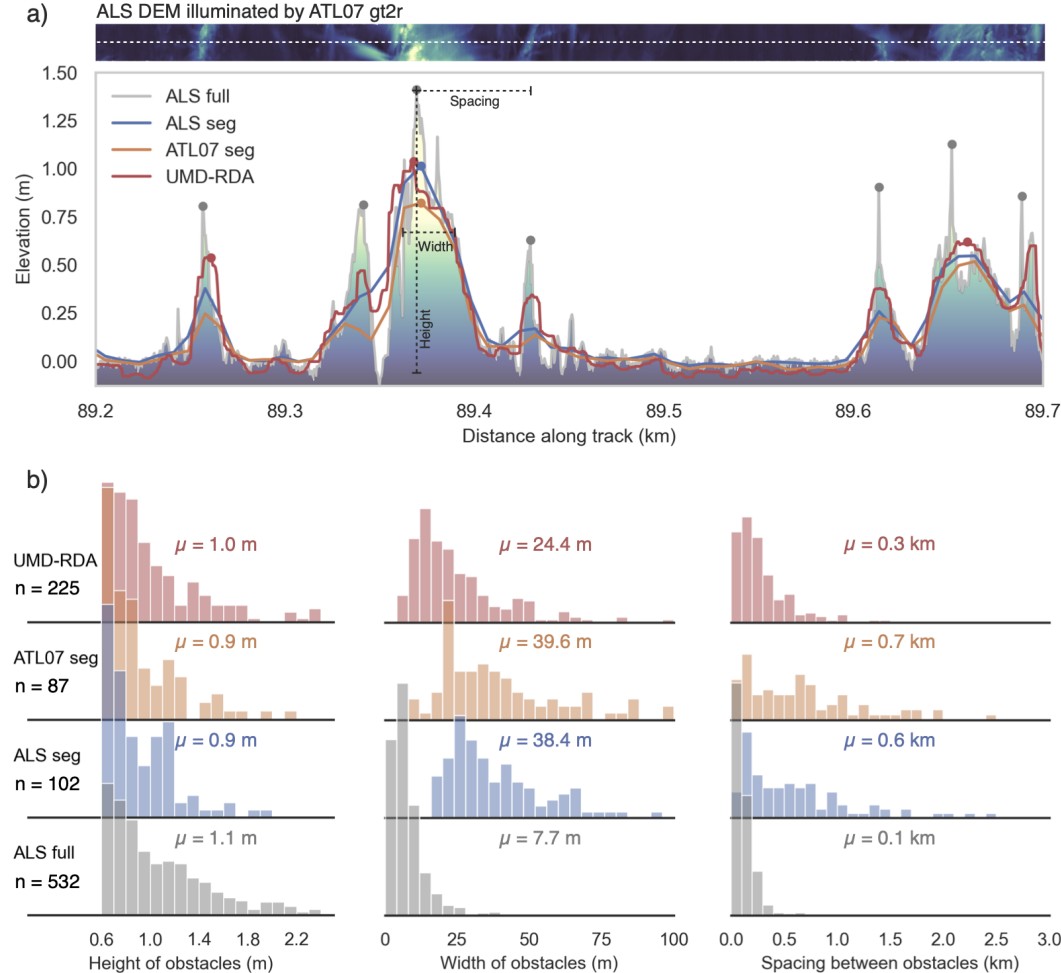

**Figure 9.** a) 500 m profile section along strong beam gt2r with sea ice elevations of different height products. Filled circles highlight peaks that exceed a height of 0.6 m with reference to local level ice. The contour in the background shows the actual topography from airborne laser scanner (ALS) data along the center line of the ALS gridded segments shown at the top, corresponding to the ATL07 segment area. b) Distributions of properties of n detected peaks (>0.6 m) along the entire helicopter profile: heights and widths of obstacles, and spacing between obstacles. $\mu$ represents the mean value for each distribution.

The same analysis is done for the weak beam gt2l (Fig. 10). Due to the longer segments, the elevation profiles of ALS seg and ATL07 seg appear even smoother compared to ALS full (Fig. 10a). In contrast, UMD-RDA is still able to resolve some of the peaks. This is also reflected in the profile statistics (Fig. 10b). In ATL07 seg and ALS seg, only 30 and 31 obstacles that exceed the 0.6 m height threshold are registered. In UMD-RDA, 223 obstacles are detected, while ALS full reveals 735 detected obstacles. Distributions for widths and spacing are similar to gt2r, but with higher mean values for ALS seg, ATL07 seg and UMD-RDA.





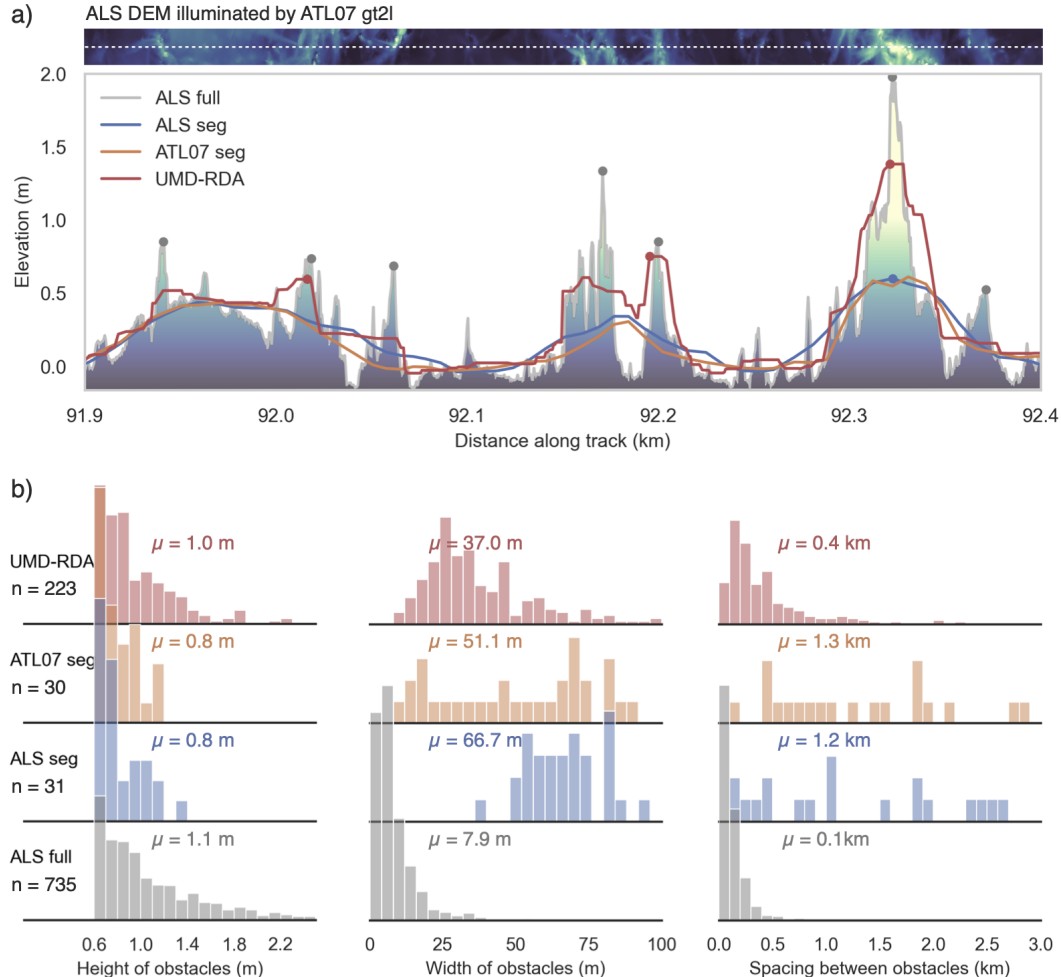

**Figure 10.** Same as Fig. 9, but for the weak beam gt2l.

## 4 Discussion

### 4.1 Differences between weak and strong beams

The differences between the weak and strong beams are a result of the surface reflectance and laser power. The laser power of
the strong beams is about four times greater than that of the weak beams. Therefore, for the data set used here, segments of
the weak beam are about 3.5 times longer to collect the 150 signal photons, while the segment spacing is comparable (Markus
et al., 2017). This results in smoother elevation profile for the weak beam (Fig. 2), while the strong beam reveals more details
of the surface topography. However, when comparing to segment-averaged ALS elevations, we find that the performance of
the weak beam is comparable to the strong beam considering correlation and rmsd (Fig. 5). Thus, weak beam elevations are
suitable for large scale studies of sea ice freeboard and thickness, when small scale topography is less important. But for the



estimation of freeboard, information about the sea level are required. What does our data set tell us about the presence of leads in ICESat-2 data?

## 4.2 Leads in ATL07

The conversion from sea ice elevation to sea ice freeboard - and subsequently to sea ice thickness - is reliant on having
observations of the local sea surface from leads in the sea ice, which must be within a reasonable vicinity of the sea ice elevation measurements (commonly between 10 - 200 km, Kwok et al. (2022)). The number of sea ice leads is an important factor in the sea ice freeboard computation and having more observations of sea surface heights from leads decreases the overall uncertainty (Di Bella et al., 2018; Ricker et al., 2016). ICESat-2, providing altimetry data of higher spatial resolution than previously seen, allows for measuring narrower leads than before, thus increasing number of estimates of the local sea
surface height. However, the requirements of having enough photons to produce one sea ice elevation measurement also applies for estimations of sea surface elevations in leads. Similarly, it is dependent on the reflectance and specularity of the surface as well as the number of detected photons. Therefore, while more leads may be observable from ICESat-2, based on the current ATL07 retrieval methodology of having to rely on 150 photons to create one segment, it is likely that not all of the leads nor the entirety of each lead will be detected. A risk that will likely be more prevalent for the weak beams where fewer photons are
detected.

Figure 8 illustrates the limitations of ATL07-identified leads. While the segment is relatively short (7 km), only one of 10 leads identified in ALS was also identified in ATL07. This contrast between ALS-identified leads and ATL07-identified leads is remarkable. From the ATL07 photon rate and surface height, we observe potential leads that were not classified as such, for example between kilometres 243 and 244. Likely, this discrepancy and non-classification is due to multiple factors. For one, the
leads observed along this profile are mostly very small, only few meters wide, refrozen cracks in the ice (Fig. 8d and e). These cracks are smaller than the ICESat-2 footprint - and much smaller than the 150-photon-aggregate segments - and therefore the elevations and photon rates get smoothed by the surrounding ice floes and do not meet the threshold criteria to be considered a lead. Additionally, ATL07-classified lead returns are expected to be specular or quasi-specular, i.e. leads with smooth surfaces, shown by the increase in photon rates in Fig. 8b (Kwok et al., 2021b). The current algorithm for ATL07 does not consider
'dark leads' (i.e. drops in photon rate) in the classification procedure (Kwok et al., 2022). The potential lead around 244 km, for example, appears to show a local minimum in photon rate, which could signal a dark lead that was not classified.

With the higher resolution of UMD-RDA, the edges of leads are more likely to be detected with higher precision due to less smoothing, which will provide more precise estimates of the width of the detected leads. ATL07 seg is likely to smooth the lead edges due to the requirement of 150-photon aggregates, leading to a larger minimum detectable width of the detected leads
in the ATL07 product, where lead detection is based on a radiometric classification Kwok et al. (2021b). But with the higher resolution of UMD also comes a detectable rougher surface in the open leads compared to the smoother ATL07 leads. Since the UMD algorithm aims to measure the top of the sea surface, there is a possibility of obtaining higher estimate of surface elevation within a lead compared to ATL07 seg. However, since UMD also aims to measuring tops of obstacles in the sea ice surface, this effect is likely to be mitigated when converting to freeboard.





Another aspect that adds to the discrepancies in Fig. 8 comes from the fact that the ALS swath is wider than the ATL07 segment width (Fig. 8d and e), and that the ALS lead finding procedure incorporates returns from outside of the overlapping segments. Future analysis of overlapping profiles that flew over more, open, and larger leads would be better to assess the ATL07 parameter lead thresholds and determine the minimum detectable width of leads. Additionally, a future modification to the ATL07 algorithm could be implemented that, for example, relaxes the 150 photon requirement for leads, as fewer signal

photons should be needed to still get an accurate height retrieval over flat surfaces.

### 4.3    How are ATL07 heights affected by surface roughness within segments?

The signal photon aggregates used to estimate surface height in ATL07 also provide information related to the surface topography within each segment. By analyzing the photon height distributions (Fig. 6), we get a sense of the roughness of the within-segment surface. While these roughness estimates, given as the standard deviation of the lognormal fit or the hist_w,

generally correlate with that from ALS (r=0.87-0.88, Fig. 5c), it is shown that they tend to overestimate the roughness of the smoothest ice and underestimate the roughness of the roughest ice (Fig. 7). This discrepancy is likely due to several reasons. Likely, the differences in surface roughness we are seeing over smooth ice in 7 are a direct result of the different impulse responses between ALS and ICESat-2. Over rougher ice, where the impulse responses would have less of an impact, it is likely that the ATL07 photon aggregation and histogram trimming play a role in the discrepancy from ALS. For a given 150 photon

segment, the height uncertainty increases as the roughness increases, which could explain some of the observed differences in Fig. 6. Additionally, the histogram trimming could remove the highest and lowest elevations from the distribution, effectively reducing its roughness, which could explain the differences in Fig. 7 at standard deviations greater than around 0.4 m.

In order to fully reconcile the retrieved within-segment elevations and roughnesses and enhance confidence in the ATL07-derived roughness, a more robust analysis involving the system impulse responses and the ATL03 photon data aggregated to

varying resolutions would be required. Additionally, future work involving the ATL07 algorithm (and specifically investigating the photon aggregation lengths, histogram trimming procedure, and the dual-Gaussian assumed surface) would be useful to better understand how to capture the sea ice topography at these scales. Until then, and due to the discrepancies in the roughness estimates, we use only the high-resolution within-segment ALS elevation measurements to estimate roughness (as opposed to the ATL07 hist_w) and to help assess the impact of roughness on the returned ICESat-2 photon distribution and retrieved

ATL07 heights.

If rougher sea ice had no impact on the retrieved ATL07 heights, we would expect Fig. 5b to show counts evenly distributed along the x-axis with no correlation, as any differences in elevation between ALS and ATL would not be related to roughness. However, there is a skewness to the distributions, with a tail that extends towards positive elevation differences at larger roughness values. These distributions indicate that over rougher sea ice, ATL07 heights tend to underestimate the surface

elevation compared to ALS. This fact can be observed in Fig. 6b and d, as the mean value of the fit lognormal distribution from ATL is less than that of ALS. It is possible that the trimming of the histograms in ATL07 could play a role. When using the non-trimmed histograms, the ATL mean values of the fitted distributions are 0.7 m and 1.15 m in b and d, respectively, which agree better with ALS. However, not trimming the histograms leads to worse lognormal fits overall, and adds potential to



include anomalous photons in the elevation retrieval (Kwok et al., 2022). Future work into the histogram trimming procedure

and dual-Gaussian assumed distribution in ATL07 would be useful to fully understand their impact on the retrieved elevations.

The lengths of the segments may also contribute to the underestimation of ATL07 heights from rougher segments, for two main reasons. First, longer segments have a higher probability of encountering obstacles that increase the roughness compared to shorter segments. This is shown in Fig. 7, where gt2l records a lower density of very-thin and level ice segments and a higher density of very rough ice segments compared to gt2r. Second, the longer segment would lead to more smoothing of

the surface obstacles in the ATL07 heights, as the single ATL07 height estimate comes from a larger area, which results in an underestimation of the highest elevations. This smoothing is observed in Fig. 9 and 10, and is more pronounced from the longer-segment gt2l data. The combination of rougher segments and more pronounced smoothing seen in gt2l segments would suggest a larger impact of roughness on the ATL07 weak beam elevations compared to the strong beam. Figure 5b confirms this hypothesis, as gt2l shows a higher correlation, indicating more of an impact, as well as a more skewed distribution with a

longer tail.

### 4.4 Mapping of ridges with ICESat-2 products

Our results show that ICESat-2 allows for detection and height estimation of individual surface topography features. However, the comparison with the high-resolution ALS data set also shows that not all ridges or obstacles will be captured. Ridge detection and sail height estimation depend on the the applied algorithm, the dimensions of the ridge, and the data product

used. In our study, we use a peak-detection algorithm with a 0.6 m height threshold referring to the surrounding level ice, similar to previous studies (e.g. Duncan and Farrell, 2022). Lowering the threshold leads to more detection, while increasing the threshold results in a lower number of detection, following an exponential or log-normal function (Fig. 4).

Given the uncertainties in the geolocation of the ATL photon heights as well as uncertainties in the drift correction of at least 0.0025 m s$^{-1}$ in x and y directions, we acknowledge potential uncertainties in our comparison due to the fact that we

consider ALS full as a reference in this study. ALS full represents the elevations along a 0.5 m wide line through the center of the ATL-illuminated area that is 13 m wide. As illustrated in Fig. 9 and 10, considering heights offsetting from the center by a few meters can lead to changes in the elevation profile and also in the detection statistics. However, since we consider a large number of points, we do not think that this affects the result of this comparison significantly.

Another aspect that is important for the ridge detection are the ridge dimensions, in combination with the along-track

resolution of the elevation data set. The segments in the ATL07 product reveal a typical spacing of 6-7 m, while the mean segment lengths are 17 m for the strong beam gt2r and 59 m for the weak beam gt2l. Therefore, narrow, but high obstacles with steep slopes, are smoothed out in the ATL07 product. In contrast, high obstacles with a plateau are better represented in ATL07. This is shown in Fig. 9, between 89.35 km and 89.4 km along track, a ridge with a width of about 30 m at mid-height is detected using the ATL07 product, while ridges with smaller dimensions are missed, for example at 89.25 km. Because of the longer

segment length, this effect is stronger for the weak beam. Eventually, the smoothing also results in an underestimation of sail heights. In contrast to ATL07, UMD-RDA only uses 5-shot aggregates, and therefore achieves a higher along-track resolution, with an average point spacing of 0.7 m (1.8 m) for the strong beam (weak beam) found in this case study. Our study shows





that using finer-resolution segments with fewer photons aggregated, such as the UMD-RDA product, can substantially improve
the ridge detection and sail height estimation over the coarser resolution segments that aggregate more photons, such as the
ATL07 product (Fig. 9 and 10). If we consider ALS full as the reference, using ATL07 results in 16 % (4 %) of detected ridges
for the strong (weak) beam. In contrast, using UMD-RDA, we obtain 42 % (30 %) of the detection number with ALS full.
Interestingly, the level of relative improvement between UMD-RDA and ATL07 is even higher for the weak beam, decreasing
by only 29 % with UMD-RDA, but by 75 % with ATL07 when referred to the results with the strong beam.

While the height distributions of the detected ridges reveal similar shapes among all products, the width distributions differ
substantially. The reason is that the width estimates strongly depend on the along-track resolution. The smoothing effect
mentioned earlier leads to an increase in width. At the same time, narrow ridges with small widths cannot be detected with
ATL07 and therefore the width distribution is biased high.

The choice of segment length (ATL07 being of varying segment length using 150-photon aggregates, whereas UMD-RDA
aims to provide observations at a per-shot basis) is also a choice made based on the overall objective of each algorithm. While
ATL07 aims to provide observations of the average local sea ice elevation, UMD-RDA aims to sample the top of the sea ice
pressure ridges. Therefore, UMD-RDA is more likely to provide higher estimates as it is based on the 99th percentile of a
trimmed 5-shot aggregate applied at per shot basis (Farrell et al., 2020). However, what is notable in this regard is how neither
ATL07 nor UMD-RDA is capable of retrieving the full extent of the surface topography, such as capturing the full height of
the obstacles or depth of the topography (to a lesser extent for the strong beam) along the transect shown in Fig. 9-10. In the
case of the weak beam data, significant smoothing across deformation features is observed in ATL07 due to the longer segment
lengths, while the UMD-RDA algorithm appears to overestimate local minima between obstacles along the transect. While it
is a function of resolution, it is also due to the UMD algorithm aiming to obtain elevation estimates using the 99th percentile
and therefore, will use the higher-elevation-photons within the aggregates as a measure of the surface elevation.

The fact that neither ATL07 nor UMD-RDA is able to capture the full extent of the surface topography likely shows the limi-
tations of ICESat-2 for specific obstacle detection. With that said, considering that ICESat-2 is a spaceborne platform observing
m-scale features from a 500 km orbit, these results are remarkable if compared to previous satellite altimeter missions.

### 4.5 Possible limitations of the study

Finally, we discuss how representative this study is, considering that our data set only covers a distance of about 130 km. The
sea ice in the surveyed area is a mix of scattered multiyear floes (e.g. the MOSAiC floe) and a larger part of first-year sea
ice (Nicolaus et al., 2022). From local observations, we know that this area was subject of several deformation events at that
time (von Albedyll et al., 2022). Although both ice types are present, neither very thick and old sea ice such as typically found
north of the Canadian Archipelago, nor large areas of very young ice <10 cm thick have been covered. The Sentinel-1 radar
image (Fig. 1a) suggests that the surveyed ice is also representative for the surrounding area. Considering the other two strong
beams, we find similar elevation distributions and standard deviations for gt1r, gt2r, and gt3r (Fig. 1b,c), indicating that gt2r
reveals a dynamic range in between gt1r and gt3r. Therefore, we conclude that our results represent sea ice as it is typical for




the Central Arctic in spring, and is also representative for the other beams. However, we note that segment lengths and spacing varies between the beams and can affect statistics.

We anticipate that over (even more) deformed and thicker sea ice, the performance differences in mapping the sea ice surface topography between UMD-RDA and ATL07 will be comparable or even higher than in our study. On the other hand, over newly
formed, rather flat thin ice, differences between UMD-RDA and ATL07 will be rather subtle.

The evaluation of signals from leads in ATL07 is limited due to the lack of larger open water leads, as we would expect them at different times of the season and other regions such as the marginal ice zone or in the Beaufort Gyre.

## 5    Conclusions

During the MOSAiC ice drift experiment, we have carried out laser scanner measurements with a helicopter, coincident with
the center beam pair of an ICESat-2 overflight in March 2020. We have processed airborne gridded sea ice surface elevations along a swath width of about 300 m, with a spatial resolution of 0.5 m, overlapping with 97 km for the strong beam gt2r and 117 km for the weak beam gt2l. This unique data set allows for studying the capabilities of ICESat-2 sea ice surface elevations in the Arctic winter period.

We have found that both the strong and the weak beam of ATL07 seg (the operational sea ice height product provided by
NASA) coincide with the corresponding segment-averaged ALS estimates (ALS seg), with correlations of 0.95 (strong beam) and 0.92 (weak beam) and root-mean-square-deviation (rmsd) of 0.07 m, which is consistent with findings in Kwok et al. (2019a). However, surface roughness is smoothed out on length scales smaller than the segment lengths. This has implications for the detection of leads, and ridges and estimates of their sail heights.

Only one lead has been identified by the ATL07 algorithm, missing especially smaller, partly refrozen, cracks that can be
seen in the ALS data set. This is a consequence of the requirement that 150 photons are needed to build a segment, which results in overlooking small leads and cracks. Aggregation of less photons for lead detection might improve the overall performance. But we also acknowledge that the ALS data set is not representative for other Arctic regions with higher lead frequency like the marginal ice zone or the Beaufort Gyre. More research is required on how lead detection can be improved, especially for small leads. Therefore, additional validation data sets and complementing measurements, such as airborne thermal infrared imaging,
would be useful.

To assess the potential of ICESat-2 data for mapping of ridges and sail heights, besides ATL07 seg, we also considered the high fidelity sea ice elevation product (Duncan and Farrell, 2022) from the University of Maryland (UMD-RDA). Here, we observe that UMD-RDA captures more obstacles with higher ridge sails more comparable to the ALS product in full resolution (ALS full), assumed to represent the true surface topography. We find that 16% (4%) of the number of obstacles in the ALS data
set are detected using the strong (weak) center beam in ATL07. Significantly higher detection rates of 42% (30%) are achieved when using the UMD-RDA product. On average for the strong beams, the obstacle sail heights are of similar magnitude (1.1 m for ALS full, 1.0 m for UMD-RDA and 0.9 m for ALS seg and ATL07 seg), whereas the width of the obstacles varies significantly. While ALS full observed a high variety in surface obstacles and the topography in high detail, neither ICESat-2



algorithm is able to capture the topography to the same extent. For the weak beams, the segments lengths of each sea ice height
segment are longer due to fewer photons being transmitted and detected, causing ATL07 missing most of the obstacle features.
Yet, our study shows that when utilizing the high-resolution of ICESat-2, demonstrated here with the UMD-RDA product, it is
possible to provide basin-scale measurements of surface roughness and sail heights, which can be used for estimation of drag
coefficients and aiding ship routing through the Arctic, if uncertainties and limitations of these products, revealed in this study,
are taken into account.

Considering the performance of the weak beam measurements, our results suggest that weak beam heights are useful for
large scale studies of sea ice freeboard and thickness, when small scale topography is less important.

ALS surveys have been carried out during the entire MOSAiC drift, providing a unique data set of sea ice surface topog-
raphy through a full seasonal cycle (Hutter et al., 2022). This study links the MOSAiC ALS measurements with ICESat-2
measurements from space, to investigate the evolution of surface topography and deformation of the sea ice near the MOSAiC
camp in the context of regional and Arctic-wide changes captured by ICESat-2.

*Data availability.* The processed helicopter laser scanner data used in this study are available under: https://doi.org/10.1594/PANGAEA.X

The sea ice concentration data product OSI-401 has been obtained from the European Organisation for the Exploitation of Meteorological
Satellites (EUMETSAT) Ocean and Sea Ice Satellite Application Facility (OSI SAF): ftp://osisaf.met.no/archive/ice/conc/, osi (2017a).
The sea ice drift data product OSI-405 has been obtained from EUMETSAT OSI SAF: ftp://osisaf.met.no/archive/ice/drift_lr/merged/, osi
(2017b). The processed Sentinel-1 image has been obtained from Drift&Noise FRAM-Sat: https://framsat.driftnoise.com. The raw Sentinel-
1 data are provided by ESA. The ICESat-2 data product ATL07 is obtained from https://doi.org/10.5067/ATLAS/ATL07.005, Kwok et al.
(2021a). The UMD-RDA product is based on ICESat-2 data product ATL03, obtained from https://doi.org/10.5067/ATLAS/ATL03.005,
Neumann et al. (2021) .

*Author contributions.* Design of the study and analysis of data: RR and SF. Planning and carrying out the helicopter flight: RR and SF. Pro-
viding ICESat-2 ground tracks during MOSAiC: NTK. Processing of the ALS data: SH, AJ, and NH. Discussion of results and conclusions:
RR, SF, RMFH, KD, SLF. Writing the manuscript: All.

*Competing interests.* The authors declare that they have no conflict of interest.

*Acknowledgements.* Helicopter data used in this manuscript was produced as part of the international Multidisciplinary drifting Observatory
for the Study of the Arctic Climate (MOSAiC) with the tag MOSAiC20192020 and the Project_ID: AWI_PS122_00.

We thank all those who contributed to MOSAiC and made this endeavour possible (Nixdorf et al., 2021). Special thanks go to the Leg III
*Polarstern* crew and the team of *Heli Service*, without their contributions this study wouldn't have been possible.



The work of SH, AJ, and NH was supported by the German Ministry for Education and Research (BMBF) project IceSense (grant 03F0866A). The work of SF was supported by the National Aeronautics and Space Administration (NASA) Cryospheric Sciences Internal Scientist Funding Model (ISFM).



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
