# Peer review of "Linking scales of sea ice surface topography: evaluation of ICESat-2 measurements with coincident helicopter laser scanning during MOSAiC"

_EGUsphere, 2022_

## Author Response (AR1)

**Reply on RC1**

Our response is highlighted in blue.

**General thoughts**

This manuscript presents a comprehensive evaluation study of ICESat-2 ATL07 as well as the high-fidelity surface elevation product developed by the University of Maryland using near-coincident helicopter-based airborne laser scanner data from MOSAiC. The detection rates, something that has only been discussed in limited capacity up till now, is very much needed to better interpret ICESat-2 topographic data. The statistics and data comparisons that the authors present are quite thorough and well communicated. Given that the paper is well-structured and the science is sound, the corrections that I point out are minimal. Overall, I think the values given here are critical in the interpretation of how to best use ICESat-2 ATL07 data on larger scales.

We thank the reviewer for these helpful comments and suggestions. We have addressed the comments and will implement the proposed changes in the paper. We have also repeated computations assuming the 11 m footprint. Please, see below our detailed response. We will provide a track-change version together with the revised paper.

**More detailed comments**

Line 11: I would remove the "the" before ICESat-2 as it is a proper name.
Agreed. We will change this accordingly.

Lines 16-17: I do not see the need to include outlook in the abstract.
Agreed. We will delete this sentence, as it is also discussed in the Conclusions section.

Line 32: I would reword "of high interest" and substitute it with something like "important" for simplicity.
Agreed. We will change this accordingly.

Line 38: A citation of a manuscript discussing how altimeter satellites "cannot resolve the surface topography" is needed here.
Agreed. We will add a recent study from Johnsen et al. (2022) where this is discussed. Moreover, we will have a reference that has investigated the effect of surface roughness in radar altimeter waveforms (Landy et al., 2020).

References:

Landy, J. C., Petty, A. A., Tsamados, M., and Stroeve, J. C.: Sea Ice Roughness Overlooked as a Key Source of Uncertainty in CryoSat-2 Ice Freeboard Retrievals, Journal of Geophysical Research: Oceans, 125, e2019JC015 820, 2020.

Johnson, T., Tsamados, M., Muller, J.-P., and Stroeve, J.: Mapping Arctic Sea-Ice Surface Roughness with Multi-Angle Imaging SpectroRadiometer, Remote Sensing, 14, https://doi.org/10.3390/rs14246249, 2022.

Line 57-58: Maybe mention why the other helicopter flights were unusable?
Agreed. We will add that other helicopter ALS surveys have not been used in this study as a direct comparison of surface features appearing in the airborne and satellite data is difficult or not possible if there is no coincidence in space and time.

Line 117: Remove ", and" from this sentence.
Agreed. We will change this accordingly.

Line 130-133: This is a bit confusing; you state that the 15th and 85th percentiles of the distribution are retained but later go on to say that the 99th percentile of the trimmed height distribution defines the sea ice surface? Is the 99th percentile of the 15-85 percentile-trimmed distribution? And if so, wouldn't that mean that it is simply the ~85th percentile of the original data suggesting it is NOT the "first interface encountered by the laser"?
We agree that this section of text is confusing. The UMD-RDA procedure to calculate sea ice height works as follows: First, a photon height distribution is constructed using a 5-shot aggregate, from which the modal height is determined. Second, photons within a range window (modal height + 10 m) and (modal height - 2 m) are retained so as to adequately capture ridge sails and leads, respectively. Third, to eliminate background (noise) photons, the height distribution is trimmed further, retaining only those photons within the 15th to 85th percentiles of the height distribution. Sea ice surface height is defined as the 99th percentile height of the final distribution. We will edit the text to add clarity to the process of defining sea ice surface height using the UMD-RDA algorithm.

Line 145: Perhaps, if it is not too complicated, it might be worth also checking the 11 m footprint? Authors can then briefly comment on how much of a change that makes to the values but do not have to include any associated plots.
We have done the computations for the 11m footprint to verify that changes are marginal. Below we show along track elevations on the left, corresponding to Figure 2b) in the paper, adding the ALS elevations using an 11 m footprint for co-registration. Moreover, we have also computed the comparison statistics using the 11m footprint (right hand figure), see Figure 5a) for comparison in the manuscript. The differences are marginal as expected. Differences for the strong beam (gt2r) are slightly higher, because segments are shorter and therefore the relative change in segment area is larger when assuming the 11m footprint.

[Figure]

[Figure]

Line 208-209: Is the local level ice subtracted from all measurements prior to the peak detection algorithm? If so, where is it discussed?
No, the peak detection is applied to the elevations after subtraction of the long-wave correction (see section 2.5). Then, in Figure 9 and 10, we refer the elevations to the level ice represented by the modal elevation over the entire data set. In fact, the caption of Figure 9 needs to be corrected, saying that elevations are referring to the modal elevation along the flight track. We will change the caption in Figure 9.

Line 319: Why do we get this discussion here? Shouldn't these differences be reported on in the Methods and data section?
In this section we want to discuss the differences between weak and strong beams in view of our findings. We acknowledge that the first part is of introductory nature. But we want to point out here that the weak beam is in fact useful if small scale roughness is not considered.

Line 356: I am not sure what "a detectable rougher surface in the open leads" could be? Despite picking out the 99th percentile of the distribution, I would expect a distribution collected over an open lead to still register as smooth.
We agree that while open leads will register as smooth compared to floes and sea ice in general, we have still observed that due to resolution differences and picking out the 99th percentile of the distribution, that UMD-RDA does indeed pick up a more varying ocean surface in the leads. An example of an ICESat-2 track during the MOSAiC campaign

(although not the track used for the comparison with ALS) with ALT07 and UMD-RDA algorithm is shown below, which aimed to investigate lead identification capabilities of the UMD-RDA and ATL07 algorithm. Here, leads identified in ATL07 were compared with elevation tracks from ATL07 and UMD-RDA.

**Sub-Fig C.**

[Figure]

Leads in ATL07 are currently identified using, but not limited to, the width of the gaussian distribution as a measure of surface roughness. Applying a quick rolling window (based on standard deviation and lower elevation measurements) to identify leads in UMD-RDA (since no UMD-RDA lead-identification algorithm is currently available) shows how elevations in the center of leads may not be identified here, because the surface here is rougher compared to the edges of the lead. This is likely caused by the resolution and algorithm of UMD-RDA, thus the points are closer and more varying compared to elevations in ATL07 - thus, a 'a detectable rougher surface within leads'. However, we shall edit the sentence for clarity since this is mainly an issue if the leads are wide.

Section 4.3: Why do you not discuss the 11 m ATL03 footprint, which is likely larger than most features you detect with the 0.5 m resolution ALS, as a source of uncertainty?
The footprint is 11 m, but the sampling distance is about 0.7 cm, which is comparable to the ALS. But indeed, the 11 m footprint adds uncertainty to the photon height, if the feature (e.g. small ridge) only takes a fraction of the footprint area, whereas the origin of the laser scanner reflection is determined very well.

Line 411: Change "previous studies" to "a previous study" or add more citations!
Agreed. We will add two more references: Hibler et al. (1972) and Tan et al. (2012).

References:

Hibler III, W. D., Weeks, W. F., and Mock, S. J.: Statistical aspects of sea-ice ridge distributions, Journal of Geophysical Research (1896-1977), 77, 5954–5970, 1972

Tan, B., Li, Z.-j., Lu, P., Haas, C., and Nicolaus, M.: Morphology of sea ice pressure ridges in the northwestern Weddell Sea in winter, Journal of Geophysical Research: Oceans, 117, 2012

Line 500-501: You've shown that the weak beams are still useful but perhaps you can elaborate on why one would use them instead or in tandem with the strong beams?
Yes, we will add a sentence here: While previous studies commonly used the strong beams (e.g. Petty et al., 2020), using weak beams in addition to derive Arctic and Antarctic sea ice freeboard and thickness maps might increase the actual area of sensed sea ice and decrease uncertainties in the gridded products because of the increased number of measurements.

**Figures**

Figure 1: "White arrows show the low resolution sea ice drift from OSI SAF" - then maybe change the appearance of the arrow indicating where the helicopter turned and the arrow indicating North?
Agreed. We will change the color of the turning arrow and the arrow indicating the North direction.

Figure 2: Change gt2l, gtr to weak beam, strong beam in the legends - which is which depends on the orientation of ICESat-2 and while it is correct for the time-frame of your study, given their mutability, I would suggest using immutable names where possible.
Agreed. We will change this.

Figure 6: What's the significance of "trimmed" and "untrimmed" here? Is the latter the version with the anomalous values? Maybe worth reiterating here.
These are the two histograms given in the ATL07 data product, where the trimmed histogram contains only photons within $2\sigma$ of the mean and is used for the fine-surface finding. The untrimmed typically can contain anomalous values. We included both to show where the ATL07 height is derived from (trimmed histogram) and also to better compare with ALS, where no height points are trimmed. We will reiterate the significance of these two histograms here and also in the data section (2.2).

Figure 7: Define the hist_w parameter again, the figures and their captions should be as independent as possible.
Agreed.

Figure 9: May I suggest further reducing the size of histograms and shifting them to the left from bottom up? This should mitigate the initial peaks completely obscuring the bars from histograms that are further up. This is especially confusing when the overlap extends to the histogram that is above the directly neighboring one.
We will revise the figure to improve readability and avoid overlap of the histograms.

**Reply on RC2**

Our response is highlighted in blue.

**General Comments**

This manuscript presents an intercomparison between different ICESat-II products (ATL07, UMD) with coincident high resolution LiDAR data captured on the MOSAiC expedition, including excellent work on relative lead detection rates. A very nice, quantitative, abstract is presented providing a concise overview. The methodology is robust and sound, and explained in an appropriate level of detail, in particular the colocation process is very well explained. A comprehensive set of results are presented and discussed extensively, leading to robust and clear conclusions, with any potential limitations clearly indicated and discussed. This is an excellent manuscript of substantive importance and requires only very minimal modifications to further enhance clarity.

We thank the reviewer for this helpful feedback. We have addressed the specific comments below. We will provide a track-change version together with the revised paper.

**Specific Comments**

Line 3: Suggest replacing 'nearly' with 'over' although I appreciate that 'nearly' was substantially correct at time of submission!
Agreed.

Line 13: Consider removing the word 'significantly' as this typically implies some kind of statistical technique has been applied, which would be quite abstract considering the ATL07 product only indicates one lead.
At line 13, the sentence "Significantly higher detection rates of 42% (30%) are achieved when using the UMD product" is related to the ridge detection (see the previous sentence). We will slightly modify the sentence for clarity: "Significantly higher detection rates" -> "Significantly higher ridge detection rates".

Line 31: Lead identification also has important anthropological implications such as on shipping/navigation, but I leave it to you to decide if including something on this would add or detract from your narrative.
Thank you for this suggestion. Shipping/navigation is a good use case of better lead monitoring. We will add a sentence in the introduction.

Line 49: This sentence introduces ambiguity because it implies that OIB has a 2m footprint, whereas I think this is only related to the ICESat-II April 2019 campaign that was flown at a higher altitude. Consider revising to refer to this specific validation campaign instead of the instrument in general.
Agreed, we revise this sentence: "... Kwok et al. (2019a) used lidar data from a campaign in spring 2019 operating at an altitude of ~1000 m, resulting in footprints of ~2 m, enough to verify the presence of  …".

Line 93: I think it would be helpful to the reader to concisely specify which interpolation method is used.
We have used a linear interpolation in 2D here. We will add this information in the manuscript accordingly.

Line 95: I appreciate reporting the altitude in ft if that was the unit it was originally measured in, but I would suggest giving preference to SI units initially.
Agreed. We will report the altitude in SI units first. But we also keep the altitude in ft since this is still commonly used in aviation.

Line 105: Make it clear these flip.
Agreed. We add a sentence to make clear that they can flip: "It must be noted that the naming `gt2l' and `gt2r' depends on the orientation of the satellite and is mutable so that 'gt2l' can be the strong beam and vice versa for other trajectories." We will also add "weak" and "strong" as annotation in Figure 2 as suggested in the other referee comment.

Line 206: This seems entirely reasonable, but could you comment at all on the stability of 250m? Ie. in the region of this value is it invariant to small changes or not.
Considering the strong beam gt2r: When we reduce the maximum distance by 10% to 225 m, we receive very similar values and counts only change for ATL07 seg (85 instead of 87 detections) and ALS seg (101 instead of 102 detections).Increasing the maximum distance by 10% leads to small changes in counts for ALS-full (534 instead of 532), ATL07 seg (89 instead of 87), and ALS seg (103 instead of 102 detections). The mean values (to one decimal place as in Fig. 9) for the different parameters remain the same.

Line 211: Really great justification!

Line 250: Suggest add 'pearson'
Agreed.

Line 327: Suggest rephrasing the rhetorical question at the end of this sentence.
Agreed. We will rephrase this sentence.

Line 437: Some characterisation of the magnitude of 'small' would be helpful here.
Agreed. We will quantify "small" here.

Line 481: Fewer photons
Agreed. We replace "less" with "fewer".

**Figures**

Figure 1 Caption: May help to instead provide a citation for framsat – similar to as you have provided with OSI SAF. This would allow you to avoid using url hyperlinks within the caption and specify the access date.
Agreed. We will add a reference for FramSat.

Figure 4 Caption: The plot order is ALS/ATL07 Seg/ALS full – but they're introduced in the caption as ATL07 Seg/ALS Seg/ALS full. Please introduce ALS seg before ATL07 seg as per plotting order.

Agreed. We will change the order in the caption.

Figure 4 Caption: I think using the word pair to refer to a set of two strong beams creates ambiguity as pair is normally a strong and a weak beam.

Actually, "pair" here refers indeed to a strong and a weak beam, of which the strong beam of the pairs number "two" and "three" is meant here. We will modify the sentence to make this clearer.

Figure 5 Caption: I would also add a sentence discussing the rmsd as in line 243, this would aid interpretation of this figure as a standalone artefact.

Agreed. We will add a sentence in the caption.